# Up-Regulation of Specific Bioactive Lipids in Celiac Disease

**DOI:** 10.3390/nu13072271

**Published:** 2021-06-30

**Authors:** Rafael Martín-Masot, Jose Daniel Galo-Licona, Natàlia Mota-Martorell, Joaquim Sol, Mariona Jové, José Maldonado, Reinald Pamplona, Teresa Nestares

**Affiliations:** 1Pediatric Gastroenterology and Nutrition Unit, Hospital Regional Universitario de Malaga, 29011 Málaga, Spain; 2Department of Experimental Medicine, Lleida Biomedical Research Institute (IRBLleida), University of Lleida (UdL), 25198 Lleida, Spain; jgalolic25@gmail.com (J.D.G.-L.); nataliamotamartorell@gmail.com (N.M.-M.); jsol.lleida.ics@gencat.cat (J.S.); mariona.jove@udl.cat (M.J.); reinald.pamplona@udl.cat (R.P.); 3Institut Català de la Salut, Atenció Primària, 25198 Lleida, Spain; 4Research Support Unit Lleida, Fundació Institut Universitari per a la Recerca a l’Atenció Primària de Salut Jordi Gol i Gurina (IDIAPJGol), 25198 Lleida, Spain; 5Department of Pediatrics, University of Granada, 18071 Granada, Spain; jmaldon@ugr.es; 6Pediatric Gastroenterology and Nutrition Unit, Hospital Universitario Virgen de las Nieves, 18071 Granada, Spain; 7Maternal and Child Health Network, Carlos III Health Institute, 28029 Madrid, Spain; 8Biomedical Research Centre (CIBM), Department of Physiology and “José MataixVerdú”, Institute of Nutrition and Food Technology (INYTA), University of Granada, 18071 Granada, Spain; nestares@ugr.es

**Keywords:** celiac disease, diacylglycerols, fatty acyls, glycerophospholipids, mass spectrometry, plasma metabolomics, steroids

## Abstract

Celiac disease (CD) is an autoimmune enteropathy linked to alterations of metabolism. Currently, limited untargeted metabolomic studies evaluating differences in the plasma metabolome of CD subjects have been documented. We engage in a metabolomic study that analyzes plasma metabolome in 17 children with CD treated with a gluten-free diet and 17 healthy control siblings in order to recognize potential changes in metabolic networks. Our data demonstrates the persistence of metabolic defects in CD subjects in spite of the dietary treatment, affecting a minor but significant fraction (around 4%, 209 out of 4893 molecular features) of the analyzed plasma metabolome. The affected molecular species are mainly, but not exclusively, lipid species with a particular affectation of steroids and derivatives (indicating an adrenal gland affectation), glycerophospholipids (to highlight phosphatidic acid), glycerolipids (with a special affectation of diacylglycerols), and fatty acyls (eicosanoids). Our findings are suggestive of an activation of the diacylglycerol-phosphatidic acid signaling pathway in CD that may potentially have detrimental effects via activation of several targets including protein kinases such as mTOR, which could be the basis of the morbidity and mortality connected with untreated CD. However, more studies are necessary to validate this idea regarding CD.

## 1. Introduction

Celiac disease (CD) is an enteropathy of autoimmune origin with a systemic and chronic trait generated by the dietary gluten and related prolamins in individuals with genetic susceptibility [1,2]. CD appears and develops from the dysfunctional interaction between genetic and environmental factors [3]. The affected subjects, usually children but also adults, display an inflammatory enteropathy with different degrees of severity [4] and clinical manifestations that range from asymptomatic to a wide variety of digestive and/or systemic signs and symptoms. Furthermore, untreated symptomatic CD patients develop relevant morbidity and mortality [5]. Thus, it can be associated to CD-related neurologic diseases; osteoporosis; T-cell lymphoma; and autoimmune endocrine diseases such as Addison’s disease, diabetes (type I), and tiroiditis [5]. Fortunately, when celiac patients are treated with a strict gluten-free diet (GFD), the clinical and histological manifestations reverse [1]. Recent studies indicate that CD affects about 1–3% of the general population in both North America and Europe [6,7,8].

The genetic predisposition of CD is linked to specific class II major histocompatibility complex (MHC) alleles. Thus, the HLA-DQ2 allele is present in more than 90% of CD subjects, whereas the DQ8 allele is present in the remaining 10% of CD patients; importantly, non-HLA genes have been described to contribute to the disease [9]. As around 40–50% of the European general population is positive for the HLA-DQ2 heterodimer [10], it is evident that additional factors are required for the onset and progression of the CD.

Alterations in metabolism seem to also be present in CD. Thus, metabolomic studies, despite being restricted to the use of NMR (nuclear magnetic resonance)-based methods, suggest lesser but significant changes in metabolites belonging to microbiome-derived metabolites, lipid metabolism, and energy metabolism [11,12,13,14,15,16]. Remarkably, differences in methionine, choline, and choline-derived lipids content in CD subjects were described, suggestive of an affectation of one-carbon metabolism. This alteration of methionine metabolism in CD subjects was recently confirmed and extended using a LC–MS/MS approach [17]. In this study, a down-regulation of the trans-sulphuration pathway in CD patients was observed, in particular.

Currently, no untargeted metabolomic analysis using LC-MS/MS or investigations regarding changes in the plasma metabolome of CD subjects have been conducted. For this reason, we designed a metabolomic study with 34 subjects including 17 CD patients and 17 healthy control siblings. The CD subjects were children under a GFD treatment. This election was based on the idea that GFD induces the return of CD signs and symptoms to a healthy state as well as induces histological changes and changes in metabolism. In addition, potential persistent metabolic alterations should be, if present, specific to the CD condition. Blood plasma was used because it is easy to obtain and is the main carrier of metabolites in the body, being well-known as the compositional profile of this biological fluid [18]. Furthermore, its composition is under strict control despite the continuous change induced by different internal and external conditions in physiological and pathological states [19,20]. These facts allow us to ensure the identification of new molecular mechanisms and targets that may lead to healthier states. The plasma metabolome profile was defined using an LC–MS/MS platform to discriminate specific phenotypic patterns linked to genotypes of CD.

## 2. Materials and Methods

### 2.1. Subjects

We performed a cross-sectional study in which 17 CD children and 17 healthy control siblings between 4 and 17 years of age were included from January to December of 2018. The study was carried out in the Gastroenterology, Hepatology, and Child Nutrition Service from the “Virgen de las Nieves” University Hospital in Granada, Spain. The diagnosis of CD was made following the European Society for Pediatric Gastroenterology Hepatology and Nutrition (ESPGHAN) criteria of 2012 [21]. Infants with liver or kidney diseases, inflammatory bowel disease, diabetes, and chronic asthma and those consuming dietary supplements containing substances with antioxidant activity were excluded. We also excluded obese patients (according to the criteria of the International Task Force) [22]. All patients signed the written informed consent document and excluded those otherwise from the study. The Ethics Committee of the University of Granada approved the study (reference number 201202400000697), and the Good Clinical Practice guidelines were followed.

### 2.2. Clinical and Sociodemographics

Participants’ clinical and sociodemographic traits, including adherence to the nutritional intervention (GFD), were evaluated by the same research team.

### 2.3. Anthropometric Measures

Anthropometric traits (height and weight) were measured in healthy control and CD subjects. Height was determined to the nearest 5 mm using a stadiometer (Seca 22, Hamburg, Germany). Body weight was determined using a mechanical balance scale (Seca200, Hamburg, Germany).

### 2.4. Blood Samples

Blood samples were obtained via venipuncture in the morning (between 07:00 and 08:00 A.M.) after fasting overnight (8–10 h) and collected in one vacutainer CPT (cell preparation tube; BD, Franklin Lakes, NJ, USA) containing sodium heparin as an anticoagulant. Blood samples were centrifuged, and plasma fractions were collected, immediately frozen in liquid nitrogen, transferred to a −80 °C freezer, and stored for 4 h for the purpose of later metabolomic analysis.

### 2.5. Untargeted Metabolomics

#### 2.5.1. Chemicals

All reagents were obtained from Sigma-Aldrich and of the highest purity available unless otherwise specified.

#### 2.5.2. Sample Processing

Plasma metabolites’ extraction was made on the basis of a previously described methodology [23]. In brief, 10 µL of plasma was mixed with 30 µL of cold methanol containing 1 μg/mL of Phe-13C as the internal standard and 1 μM BHT as the antioxidant. After incubation at 4 °C for 60 min, the samples were centrifuged at 12,000× *g* for 3 min. Then, the supernatant was filtrated (CLS8169, Sigma, Madrid, Spain) and transferred to vials with glass inserts (Agilent, Barcelona, Spain) for chemical analysis. Samples were decoded and randomized before the LC-MS analysis. To serve as quality control for metabolite extraction, a pool of plasma samples with internal Phe-13C was used and injected every 5 samples.

#### 2.5.3. Analysis Conditions

Two microliters of metabolic extract were analyzed according to a previously described method [23]. A reversed-phase column (Zorbax SB-Aq 2.1 × 50 mm, 1.8 µm particle size, Agilent Technologies, CA, USA) with a pre-column (Zorbax SB-C8 Rapid Resolution Cartridge, 2.1 × 30 mm, 3.5 µm particle size, Agilent Technologies, CA, USA) set at 60 °C was used for chromatographic separation [23]. Afterwards, electrospray ionization was made in positive and negative ion modes using N_2_ at a pressure of 50 psi for the nebulizer with a flow of 12 L/min and a temperature of 325 °C.

The analysis was performed using a UHPLC 1290 model coupled to a Q-TOF 6545 instrument (Agilent Technologies, CA, USA) as previously described [23], and data were collected using the MassHunter Data Analysis Software (Agilent Technologies, CA, USA).

### 2.6. Data Analyses

Mass Hunter Qualitative Analysis Software (Agilent Technologies, Barcelona, Spain) was used to obtain samples’ molecular features, representing different co-migrating ionic species of a given molecular entity using the Molecular Feature Extractor (MFE) algorithm (Agilent Technologies, Barcelona, Spain). The Mass Hunter Mass Profiler Professional Software (Agilent Technologies, Barcelona, Spain) was used to make an untargeted metabolomic analysis over the extracted features. Only features with a minimum of 2 ions were selected. Then, the molecular traits in the samples were aligned using a retention time window of 0.1% ± 0.25 min and 30.0 ppm ± 2.0 m Da. Only features found in at least 70% of the QC samples and with less than 20% of relative standard deviation among QC samples were considered to correct for individual bias. Instrumental drift was corrected using a LOESS approach [24,25]. Multivariate statistics (Principal Component Analysis (PCA), Partial Least Squares-Discriminant Analysis (PLSDA), and Hierarchical and Classification Analyses) were performed using the Metaboanalyst software [26,27].

Statistical analyses were conducted using R version 3.6.1. [28] A paired-t-test was used to evaluate sibling differences in clinical and MS comparisons between groups. The minimum signification level was set at *p* < 0.05 without multiple-testing adjustment, and Benjamini–Hochberg’s False Discovery Rate (FDR) p-values are reported as complementary information. Heatmaps and forest plots were performed using the heatmap and forest plot packages, respectively [29,30].

Features were defined by the exact mass and retention time using HMDB [31] (accuracy < 30 ppm) and PCDL database (Agilent Technologies, Barcelona, Spain), which uses retention times in a standardized chromatographic system as an orthogonal searchable parameter to complement accurate mass data (accurate mass retention time approach) [32]. Finally, identities were confirmed by MS/MS by checking the MS/MS spectrums using HMDB [31].

## 3. Results

### 3.1. Clinical Traits

The baseline clinical traits of the study population are shown in Table 1. A total of 34 children were included in the study, 17 CD and 17 healthy siblings. There were no differences between groups, including factors that could affect metabolomics’ profile such us dietary habits and physical activity.

### 3.2. Untargeted Metabolomics

With the aim to analyze the effect of the celiac condition in the whole circulating metabolome, an untargeted metabolomic analysis was performed. Baseline correction, peak picking, and peak alignment were determined based on acquired data, resulting in a total of 48,952 molecules from both ionization modes (positive and negative). After quality control assessment, filtering, and correction of the signal, 4893 features remained that were used for statistical analysis. Multivariate statistics suggest that there is not a specific metabolomic signature associated with the celiac condition (data not shown), but when we searched for specific celiac biomarkers, we found 209 molecules statistically different between groups and five of them (lithocholyltaurine, 18-Oxocortisol, 5alpha-Pregnan-3alpha, 20beta-diol disulfate, and an unidentified compound, all up-regulated in CD individuals) retained their significance after applying Benjamini–Hochberg’s FDR (Appendix A). Among all of the significant molecules, 36 of them were annotated (Figure 1 and Figure 2), wherein 23 (64%) were increased and 13 (36%) decreased in CD patients. Specifically, we described two carboxylic acids and derivates increased in CD, four fatty acyls, five glycerolipids, eleven glycerophospholipids, one organoxigen compound, and two sphingolipids were affected by the CD condition. Furthermore, among 36 molecules with a potential identity, eight of them were lipid species belonging to steroid metabolism and two were involved in bilirubin metabolism. Interestingly, nine of these identified molecules have been involved in cell signalling pathways (diacylglycerides (DG), lysophospholipids, and ceramides) and are mainly increased in CD subjects.

## 4. Discussion

The variability in clinical manifestations of CD is due to the interindividual differential vulnerability supported by genetic, immunological, and environmental factors. Regarding the diversity of signs and symptoms generated by CD, recent observations suggest that metabolic alterations in subjects with CD must also be considered. Although in most cases nowadays the diagnosis of CD requires confirmation by a positive biopsy, a better understanding of the metabolic processes underlying this pathology could offer the opportunity to uncover potential new physiopathological mechanisms and biomarkers useful for the diagnosis of doubtful cases due to the histological results as well as for follow-up of the disease.

Effectively, available evidence from metabolomic studies certified the presence of minor but significant differences in the metabolic profile between healthy individuals and celiac patients (children and adults) [11,12,13,14,15,16,17]. In these studies, using NMR or MS-based methods, biological samples from different origins including faecalis material, plasma, serum, and urine were analyzed. Overall, the metabolomic changes described are associated with modifications in gut microflora and/or intestinal permeability, malabsorption, and alterations in metabolism [11,12,13,14,15,16,17]. Thus, lower concentrations of amino acids (asparagine, isoleucine, methionine, proline, and valine); diverse metabolites (choline, creatinine, lactate, lipids (phosphatidylcholines), methylamine, methylglutarate, and pyruvate); and higher levels of 3-hydroxybutyric acid, glucose, and triacyl glycerides were the main differences observed in the serum metabolome between CD subjects and healthy controls [11,12,13,14,15,16]. It is also particularly remarkable that the decreased levels of choline, methionine, and phosphatidylcholines [15,16] along with the decreased content of metabolites in the trans-sulphuration pathway [17] clearly suggest defects in the one-carbon metabolism in CD. Importantly, these studies also verified that the introduction of a strict GFD seems to quasi completely reverse the metabolic profile of CD subjects to the healthy condition.

In the present study, our findings demonstrate the persistence of alterations in the whole circulating metabolome of CD patients in spite of dietary intervention. These metabolomic changes are, however, minor as only 209 metabolites seem to be affected, which represent only around 4% of the total plasma metabolome analyzed. Unfortunately, among these 209 metabolites, only 36 were identified, with 23 (64%) increased and 13 (36%) decreased in CD. Interestingly, we identified that molecules mainly belong to lipid families and are involved in cell signaling (DG, lysophospholipids, and ceramides). These lipid species are mostly increased in CD. Furthermore, we found fatty acid derivates involved in inflammatory pathways such as leukotrienes and thromboxanes. Remarkably, eight of the lipid species identified belonged to steroid metabolism and two of them were involved in bilirubin metabolism, suggesting potential alterations at the hepatic level as well as in the endocrine system and, in particular, the adrenal gland. Although it has been described that endocrine manifestations occur in celiac disease [34], this is the first time that potential alterations affecting the functioning of the adrenal gland are reported. More studies, however, are necessary to explore this idea in CD.

Among the observed changes in lipid molecular species, the increased content of DG (three species) and phosphatidic acid (PA) (one species) is particularly relevant and requires special attention. DG has exclusive functions as a constituent of cell membranes, intermediate in lipid metabolism and lipid-mediated signaling, and their levels are strictly regulated [35]. DG signals are particularly relevant in neuronal and immune tissues, playing a central role in the control of neuron communication, phagocytosis, and the control of immune responses [36]. Simply put, DG is generated by hydrolysis of phosphatidylinositol 4,5-bisphosphate, which is also the substrate for phosphatidylinositol 3 kinase activity. DG is phosphorylated in a reaction catalyzed by DG kinases to generate PA, the first step in a series of reactions that replenish phosphatidylinositol 4,5-bisphosphate levels in the phosphatidylinositol cycle. Other DG sources include the pool generated from PC (phosphatidylcholine) and ceramides [36]. Importantly, DG and PA, as lipid second messengers, play important roles in regulating several target proteins including numerous protein kinases such as mTOR [37], among several others. Interestingly, a sustained inflammatory response mediated by the activation of mTOR in CD has been recently described [38]. Therefore, our findings suggest potential defects of these biological processes subsequently to the up-regulation of DG metabolism and for which the consequences should be evaluated in CD patients.

Regarding limitations, the present study’s sample size was relatively small, and consequently, the results must be interpreted with caution. P-values were not adjusted for multiple testing; thus, the results should be further validated with complementary techniques. The fact that five of the celiac children were on a GFD for less than 12 months could influence the results, considering a long-term GFD could act on the metabolic pathways after 12 months, as other studies demonstrated within an adult population [11]. However, this is the first sibling-based metabolomic approach in pediatric CD. Moreover, the fact that study participants were matched with their siblings reduces the differences that can occur in metabolomic processes [39], which could be influenced by several factors, thus decreasing the variability. Another limitation that should be considered is that glycerophospholipid metabolism could not be fully assessed because the metabolite extraction method used is thought to extract more polar components and triacyl glycerides were not extracted and detected. Consequently, the potential role of these molecules as bioenergetics sources has been dismissed.

In conclusion, our results suggest the persistence of minor but significant alterations in the lipid metabolism of CD subjects, despite being under a GFD, that can be an expression of a new biological mechanism underlying the CD pathology, which requires further studies to be verified.

## Figures and Tables

**Figure 1 nutrients-13-02271-f001:**
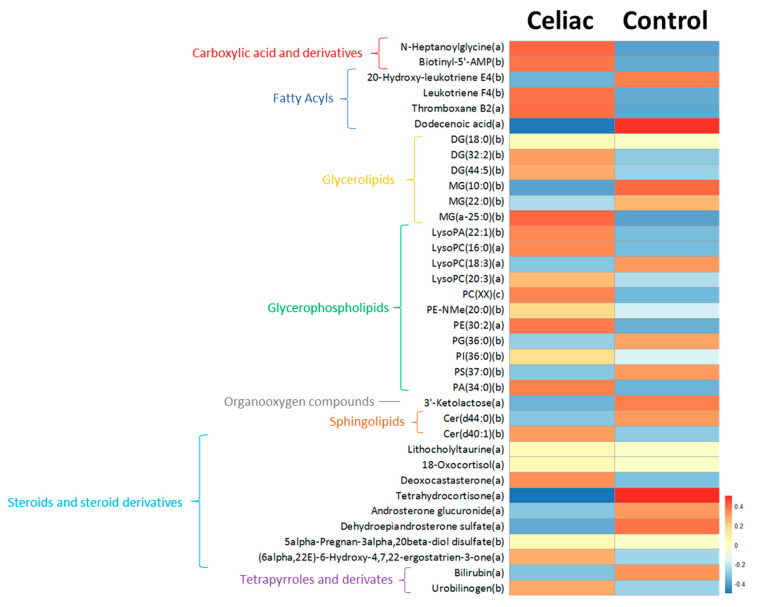
Putatively annotated plasma metabolites significantly different in CD subjects compared to healthy siblings. Levels of statistically significant molecules are represented using a heatmap. Each line of the heatmap represents a lipid species colored by its abundance intensity normalized with an internal standard, log-transformed, and auto-scaled. The scale from blue to red indicates the normalized abundances in arbitrary units. All compounds are putatively annotated based on physicochemical properties and/or spectral similarity with public/commercial spectral libraries [33]. (a) ID based on exact mass, RT, and MS/MS spectrum; (b) ID based on exact mass and RT; and (c) ID based on MS/MS spectrum. PC(XX): the specific phosphatidylcholine species could not be annotated, but its MS/MS spectrum displayed a PC-characteristic pattern.

**Figure 2 nutrients-13-02271-f002:**
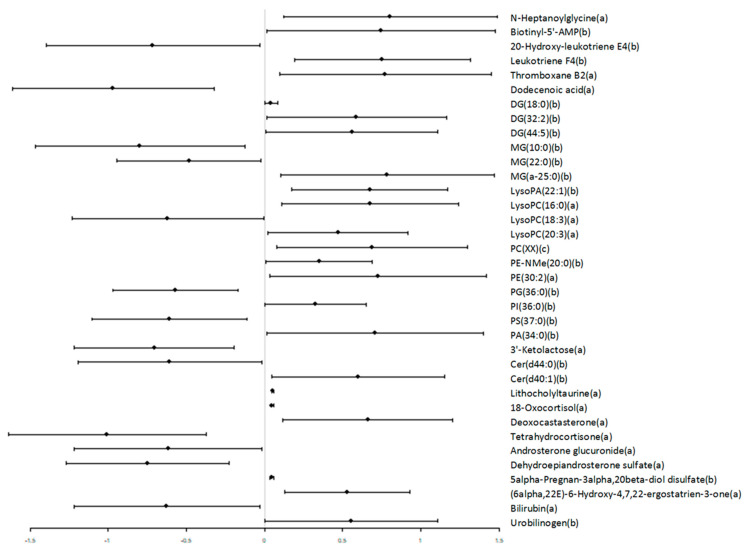
Lipid metabolism seems to be preferentially affected by CD. Levels of statistically significant molecules are represented using a forest plot. Forest plot dots represent the estimated difference in means, and the whiskers represent the confidence intervals. All compounds are putatively annotated based on physicochemical properties and/or spectral similarity with public/commercial spectral libraries [33]. (a) ID based on exact mass, RT, and MS/MS spectrum (high fiability); (b) ID based on exact mass and RT; and (c) ID based on MS/MS spectrum. PC(XX): the specific phosphatidylcholine species could not be annotated, but its MS/MS spectrum displayed a PC-characteristic pattern.

**Table 1 nutrients-13-02271-t001:** Demographic and biological characteristics of the study population.

Trait	Healthy Siblings (*n* = 17)	Celiac Children (*n* = 17)	*p*-Value
Sex (female, n (%))	10 (58.8)	13 (76.4)	0.271
Age (years)	11.25 (4.23)	9.39 (2.77)	0.145
Weight (kg)	38.54 (16.9)	30 (9.63)	0.082
Height (cm)	140.66 (20.32)	131.2 (19.56)	0.178
BMI (kg/m^2^)	18.5 (3.98)	17 (1.58)	0.166
Mediterranean diet (MD) adherence (n (%))			
*Low*	1 (5.9)	1 (5.9)	
*Medium*	8 (47.1)	7 (41.2)	0.936
*High*	7 (41.2)	8 (47.1)	
Gluten-free diet (GFD)	0	17	
Moderate physical activity (min/week)	69.64 (37)	81.4 (56.9)	0.52
HLA DR-DQ genotype			
*Negative*	5	0	
*HLA-DQ2+*	11	14	
*HLA DQ8+*	0	0	
*HLA-DQ2 + DQ8+*	1	3	

Values shown as mean (standard deviation) unless otherwise indicated.

## Data Availability

The data presented in this study are available on request from the corresponding author.

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
