# Peer review of "Up-Regulation of Specific Bioactive Lipids in Celiac Disease"

_nutrients, 2021, doi:10.3390/nu13072271_

Round 1

Reviewer 1 Report

The authors investigated plasma metabolomics changes in the case of celiac disease in children. The manuscript is well written but contains several missing issues should be improved in order to help the reader in the understanding of the work.

General text remarks:

There are numerous missing gaps between the words. For instance, in the abstract lines 23, 28, 34. Please correct it throughout the text.

Untargeted metabolomics:

2.5.2. Sample processing.

Please explain why the internal standard was added to the plasma samples and how it was used further. Was it used to determine the absolute concentrations of the metabolites?

2.5.3. Analysis conditions

In the description of the LC-MS instrument, I suggest to use the generally well known and accepted abbreviations and avoid repetitions (for example, electrospray ionization is cited 3 times in the lines 135 to 140).

I suggest: The analysis was performed using UHPLC 1290 model coupled to LC-QTOF 6545 instrument (Agilent).

Please use the uniform description of products from Agilent Technologies throughout the text (Spain and the USA origins are both indicated for the products, whereas I’m not sure that their LC-MS systems are made in Spain).

Line 143:

How did you use the QC samples further? Were they used for the statistical analysis?

2.6. Data analysis.

Please explain the statistical methods applied to the MS data. What was the multivariate analysis (even if the results are not shown)?

The authors say: Paired T-test was used to measure sibling differences. Was the T-test used only for the data of general biological characteristics of study participants, as indicated in the Table 1? Was the same test applied to the MS data to determine the differences between the groups?

Results

3.2. Untargeted metabolomics.

Line 160: The authors write “after quality assessment”…. Please explain what this quality assessment includes.

Line 162: The authors write: multivariate statistics reinforced the idea that there is no specific metabolomics signature associated with celiac condition. Why did it reinforce? Was the absence of the differences previously shown for CD versus healthy subjects?

What type of statistical analysis permitted to differentiate the 209 molecules statistically different between the groups? What does “Dataset” mean?

It is curious that multivariate statistical analysis did not permit to distinguish between the groups, whereas there are 209 metabolites that differ significantly. Could it be due to the fact that when analyzing a huge number of variables, the most interesting variables were “diluted” and thus were not detected?

Could the authors explain what type of chemometrics analysis was applied to their data and how the data were pre-processed. Did you take into consideration, for example, the blank samples and QC samples to confirm that each variable included in the statistical analysis did not correspond to the “noise”? 

The fact that multivariate statistical analysis did not show the differences between the groups indicates that the groups are very similar. How could you explain that a such a large number of metabolites (209) are significantly altered between the groups? Did you use the method of “multiple testing”? In this case, if submitting 4893 features to this analysis and accepting the 5% confidence level, there are the chances to find about 5% of compounds (and it corresponds to 244 features, not so far from 209) that may seem to be significantly different, just because there is less than 5% of chance to find the difference. I suggest that complementary models should be used to confirm the differences about groups.

Line 166: Were the 36 annotated metabolites identified as compared to standard compounds?

Figure legends 1 and 2:

Please use Metabolomics Standards Initiative levels (MSI, Sumner et al., Metabolomics. 2007 Sep; 3(3): 211–221.) to describe the identification levels of the metabolites.

What are the differences between a and b identification? Both are based on exact mass, RT (as compared to a standard compound?), MS/MS spectrum, but a) is of “high fiability” and b) of “medium fiability”. Please explain this.

Discussion:

I have a general question for your study: as I understood, you included in your study the siblings with or without CD disease. Were the CD-subjects in the remission state? Were they demonstrating the acute symptoms of the disease when sampling? If they follow the gluten free diet, it may improve their state, and thus diminish the possible differences that could be observed at the metabolite level. I this is the case, I think it should be noted as the limitation of the study and included in the discussion

Author Response

We would like to thank to Reviewer 1 the effort and time devoted to our manuscript. Thank you for the constructive comments that obviously have improved our manuscript.

Comment 1:

General text remarks:There are numerous missing gaps between the words. For instance, in the abstract lines 23, 28, 34. Please correct it throughout the text.

Answer 1:

Thank you. It has been carefully revised.

Comment 2:

Please explain why the internal standard was added to the plasma samples and how it was used further. Was it used to determine the absolute concentrations of the metabolites?

Answer 2:

We thank the reviewer for the comment.

Internal standard was used to monitor potential equipment drifts and stablish a correct retention time and accurate mass windows. Furthermore, relative abundance of internal standard of each sample was used to normalize the signal after injection.

We clarify this point also in the new version of the manuscript.

Comment 3:

In the description of the LC-MS instrument, I suggest to use the generally well known and accepted abbreviations and avoid repetitions (for example, electrospray ionization is cited 3 times in the lines 135 to 140).

I suggest: The analysis was performed using UHPLC 1290 model coupled to LC-QTOF 6545 instrument (Agilent).

Answer 3:

We have modified the text according to the reviewer’s suggestion.

Comment 4:

Please use the uniform description of products from Agilent Technologies throughout the text (Spain and the USA origins are both indicated for the products, whereas I’m not sure that their LC-MS systems are made in Spain).

Answer 4:

We have homogenized the text according to the reviewer’s comment.

Comment 5:

Line 143:How did you use the QC samples further? Were they used for the statistical analysis?

Answer 5:

We apologize for not including this information in the previous version of the manuscript. The QC samples were used for quality assessment. We have added all the information in data analyses section:

…Only features found in at least 70 % of the QC samples and with less than 20% of relative standard deviation among QC samples were taken into account to correct for individual bias.Instrumental drift was corrected using a LOESS approach…”

Comment 6:

Please explain the statistical methods applied to the MS data. What was the multivariate analysis (even if the results are not shown)?

Answer 6:

According to reviewer’s request we have added the information of statistical methods used in the new version of the manuscript:

“Multivariate statistics (Principal Component Analysis (PCA), Partial Least Squares - Discriminant Analysis (PLSDA), and Hierarchical and Classification Analyses) were performed using Metaboanalyst software”

Comment 7:

The authors say: Paired T-test was used to measure sibling differences. Was the T-test used only for the data of general biological characteristics of study participants, as indicated in the Table 1? Was the same test applied to the MS data to determine the differences between the groups?

Paired T-tests were also used when assessing MS differences between groups. We have specified that they were used for all the analyses in the new version of the manuscript:

“…Paired-T-test was used to measure sibling differences in clinical and MS comparisons between groups…”

Comment 8:

Line 160: The authors write “after quality assessment”…. Please explain what this quality assessment includes.

Answer 8:

We have added this information in the methods section of the new version of the manuscript:

…Only features found in at least 70 % of the QC samples and with less than 20% of relative standard deviation among QC samples were taken into account to correct for individual bias.Instrumental drift was corrected using a LOESS approach…”

Comment 9:

Line 162: The authors write: multivariate statistics reinforced the idea that there is no specific metabolomics signature associated with celiac condition. Why did it reinforce? Was the absence of the differences previously shown for CD versus healthy subjects?

Answer 9:

We thank the reviewer for the comment. We have changed reinforced for suggested in order to avoid misunderstanding.

Comment 10:

What type of statistical analysis permitted to differentiate the 209 molecules statistically different between the groups? What does “Dataset” mean?

Answer 10:

As it is explained in Methods Section, Data Analyses, Paired T-Test is applied to obtain the statistically differential molecules (p<0.05).

Dataset is a commonly used excel file where all the relevant information about the metabolites is included. We have specified that it is a supplementary material.

Comment 11:

It is curious that multivariate statistical analysis did not permit to distinguish between the groups, whereas there are 209 metabolites that differ significantly. Could it be due to the fact that when analyzing a huge number of variables, the most interesting variables were “diluted” and thus were not detected?

Answer 11:

We thank the reviewer for this comment. It is not surprising in this kind of metabolomic analyses not finding clear differences when multivariate statistics is applied but finding specific differential metabolites between groups. This happens when there are other important factors (age, gender, diet, habits, risk factors, etc.) that affects more the metabolome than the factor of study (CDin this case).

Comment 12:

Could the authors explain what type of chemometrics analysis was applied to their data and how the data were pre-processed. Did you take into consideration, for example, the blank samples and QC samples to confirm that each variable included in the statistical analysis did not correspond to the “noise”? 

Answer 12:

We thank the reviewer for this comment. We have included all this information in the Data analyses subsection in the new version of the manuscript.

Comment 13:

The fact that multivariate statistical analysis did not show the differences between the groups indicates that the groups are very similar. How could you explain that a such a large number of metabolites (209) are significantly altered between the groups? Did you use the method of “multiple testing”? In this case, if submitting 4893 features to this analysis and accepting the 5% confidence level, there are the chances to find about 5% of compounds (and it corresponds to 244 features, not so far from 209) that may seem to be significantly different, just because there is less than 5% of chance to find the difference. I suggest that complementary models should be used to confirm the differences about groups.

Answer 13:

As we have stated before, regardless of the number of differential metabolites, there may be other factors that might have a wider effect to the metabolome than CD.

Since it was an exploratory analysis and we recruiteda relatively low number of participants we have reported all the significant metabolites without adjusting the p-values. However, we have specified the FDR p-values in the supplementary DataSet, and we have stated that 4 of the metabolites were significant when adjusting for Benjamini-Hochberg’s false discovery rate in the new version of the manuscript:

“…when we searched for specific celiac biomarkers we found 209 molecules statistically different between groups, and 4 of them (Lithocholyltaurine, 18-Oxocortisol, 5alpha-Pregnan-3alpha,20beta-diol disulfate and an unidentified compound, all of them upregulated in CD individuals) kept their significance after applying Benjamini-Hochberg’s FDR  (Supplementary DataSet)…”

We have also stated this in the limitations section, as follows:

“…P-values were not adjusted for multiple testing, so the results should be further validated with complementary techniques…”

Comment 14:

Line 166: Were the 36 annotated metabolites identified as compared to standard compounds?

Answer 14:

We apologize for not including this information in the manuscript. We have added all this information in the new version of the manuscript (Methods, Data Analyses subsection”.

“Features were defined by exact mass and retention time using HMDB (accuracy < 30 ppm) and PCDL database (Agilent Technologies, Barcelona, Spain), which uses retention times in a standardized chromatographic system as an orthogonal searchable parameter to complement accurate mass data (accurate mass retention time approach). Finally, identities were confirmed by MS/MS by checking the MS/MS spectrums using HMDB”

Comment 15:

Figure legends 1 and 2: Please use Metabolomics Standards Initiative levels (MSI, Sumner et al., Metabolomics. 2007 Sep; 3(3): 211–221.) to describe the identification levels of the metabolites.

Answer 15:

We thank the reviewer for the comment. We have introduced the information of the paper requested in the new version of the manuscript.

“All compounds are putatively annotated compounds based upon physicochemical properties and/or spectral similarity with public/commercial spectral libraries”

Comment 16:

Figure legends 1 and 2:What are the differences between a and b identification? Both are based on exact mass, RT (as compared to a standard compound?), MS/MS spectrum, but a) is of “high fiability” and b) of “medium fiability”. Please explain this.

Answer 16:

The difference is the confidence of the MS/MS spectrum. Public databases not always offer the experimental data and sometimes there are only predicted data available. Furthermore, not all the MSMS spectrum is equally “clean” and the matches found between sample compounds and database compounds are not always the same. If we don’t have any doubt of the confidence we assign a high fiability. In the other cases we assign medium fiability.

Comment 17:

Discussion: I have a general question for your study: as I understood, you included in your study the siblings with or without CD disease. Were the CD-subjects in the remission state? Were they demonstrating the acute symptoms of the disease when sampling? If they follow the gluten free diet, it may improve their state, and thus diminish the possible differences that could be observed at the metabolite level. I this is the case, I think it should be noted as the limitation of the study and included in the discussion

Answer 17:

Thank you for your comment. All the children were asymptomatic, with adequate growth and normal serology (except for 1 patient). Of the 17 children, 12 were on a gluten-free diet for more than 18 months, 5 for a little less than 12. This point has been incorporated in the text as a limitation: “The fact that 5 of the celiac children were on a GFD for less than 12 months could influence the results, since a long-term GFD could act on the metabolic pathways after 12 months, as other studies showed in an adult population [11]”, but the fact that in our study we could demonstrate the persistence of alterations in the whole  circulating metabolome of all CD patients, could mean that the changes are independent of the time on a gluten-free diet. 

Reviewer 2 Report

In the present original paper Martin-Masot et al demonstrated that, in a group of 17 children with celiac disease (CD), despite a gluten free diet (GFD), metabolomic analysis showed an impaired metabolic profile, with alteration of lipid species (glycerolipids and fatty acyls). Main comments:

1) I feel that discussion is not fully focused. Indeed, Authors did not give a possible pathogenetic explanation for their findings and they did not compare their results with similar studies in literature.

2) Page 3 lines 102-103: Authors stated to have assessed GFD adherence, however in the Results section they did not specify whether adherence was satisfactory for all children. Could incomplete adherence have influenced the results? Moreover, which tool was used to evaluate adherence?

3) Please consider to rephrase the title: it is hard to understand

Author Response

We would like to thank to Reviewer 2 the effort and time devoted to our manuscript. Thank you for the constructive comments that obviously have improved our manuscript.

Comment 1:

I feel that discussion is not fully focused. Indeed, Authors did not give a possible pathogenetic explanation for their findings and they did not compare their results with similar studies in literature.

Answer 1:

We appreciate the reviewer's comment and share the idea that it would be interesting to expose some potential pathogenetic mechanism. We understand that the alterations that we have observed that affect various bioactive lipids represent per se a potential pathogenetic mechanism of which we do not know whether it is at the origin of the process or develops after it. For this reason, we indicate the importance of carrying out more studies to assess the real meaning of the observed changes. In the light of current knowledge about celiac disease, discerning the biological significance of the changes found would be excessively speculative and, therefore, we ask the reviewer to allow us to continue the discussion in the terms in which we present it.

Comment 2:

Page 3 lines 102-103: Authors stated to have assessed GFD adherence, however in the Results section they did not specify whether adherence was satisfactory for all children. Could incomplete adherence have influenced the results? Moreover, which tool was used to evaluate adherence?

Answer 2:

Thank you for your comment. There is no a validated tool for measuring gluten free diet adherence in Spanish children, so dietary intake was assessed through a three-day food record, two on weekdays and one on the weekend. For each meal, participants were requested to report an exhaustive description of the food and the recipes (including cooking methods and sugar or fats added during the meal preparation), food amount and the brands of packaged foods consumed. All diaries were analyzed by the same trained dietitian.

All the children were asymptomatic, with adequate growth and normal serology (except for 1 patient). Of the 17 children, 12 were on a gluten-free diet for more than 18 months, 5 for a little less than 12. This point has been incorporated in the text as a limitation: “The fact that 5 of the celiac children were on a GFD for less than 12 months could influence the results, since a long-term GFD could act on the metabolic pathways after 12 months, as other studies showed in an adult population [11]”, but the fact that in our study we could demonstrate the persistence of alterations in the whole  circulating metabolome of all CD patients, could mean that the changes are independent of the time on a gluten-free diet.

Comment 3:

Please consider to rephrase the title: it is hard to understand.

Answer 3:

In accordance with the reviewer’s comment, we have modified the title of our manuscript.

Round 2

Reviewer 1 Report

The authors have improved the manuscript, namely by adding the methodological information.

I still have few comments:

Missing gaps between the words: may be due to the transformation to the pdf (?), but, for example, in the section 2.6 Data analysis there numerous missing gaps.

Then, line 161, it is written “pheatmap” (I imagine “heatmap”)

Line 192, in the word “specifically”, “a” is missing

Line 192, in “2 carboxylic acid”, I think, “s” is missing in acids.

Please check carefully the text for these misprints.

In Figure legend 1, I suggest not to use the term “identified plasma metabolites”, but directly use “putatively annotated”. Even if the authors add this information further, I find that the figure legend needs to be clear for the reader.

As the authors consider, using only the comparison with HMDB MS-MS spectral information, mainly based on theoretical fragmentation and not confirmed by the experimental work, does not permit the identification of compounds. Moreover, the description between “high fiability” and “medium fiability” is rather arbitrary, and, in my opinion, will not be clear for the readers, especially for the readers without the strong experience in analytical chemistry approaches.

Author Response

The authors have improved the manuscript, namely by adding the methodological information.I still have few comments:

Comment 1:

  • Missing gaps between the words: may be due to the transformation to the pdf (?), but, for example, in the section 2.6 Data analysis there numerous missing gaps.
  • Then, line 161, it is written “pheatmap” (I imagine “heatmap”)
  • Line 192, in the word “specifically”, “a” is missing
  • Line 192, in “2 carboxylic acid”, I think, “s” is missing in acids.

Answer1:We have modified the text according to the reviewer’s suggestion.However “pheatmap” is not an error, this is the name of the package.

Comment 2:  In Figure legend 1, I suggest not to use the term “identified plasma metabolites”, but directly use “putatively annotated”. Even if the authors add this information further, I find that the figure legend needs to be clear for the reader.

Answer 2: Thank you. We have modified the text according to the reviewer’s suggestion.

Comment 3:  As the authors consider, using only the comparison with HMDB MS-MS spectral information, mainly based on theoretical fragmentation and not confirmed by the experimental work, does not permit the identification of compounds. Moreover, the description between “high fiability” and “medium fiability” is rather arbitrary, and, in my opinion, will not be clear for the readers, especially for the readers without the strong experience in analytical chemistry approaches.

Answer 3:We thank the reviewer for this comment. We have modified the text and the figures according to the reviewer’s suggestion.

Reviewer 2 Report

Answers were satisfactory

Author Response

Thank you for your comments that obviously have improved our manuscript.